# GeoCross: A Privacy-Preserving and Fine-Grained Authorization Scheme for Cross-Chain Geological Data Sharing

**DOI:** 10.3390/s25247625

**Published:** 2025-12-16

**Authors:** Licheng Lin, Bin Feng, Pujie Jing

**Affiliations:** 1Institute of Geophysical and Geochemical Exploration, Chinese Academy of Geological Sciences, Tianjin 300300, China; guocheng1016@gmail.com; 2State Key Laboratory of Deep Earth Exploration and Imaging, Tianjin 300300, China; 3North China Institute of Aerospace Engineering, Langfang 065000, China; pujiejing1996@nciae.edu.cn

**Keywords:** cross-chain, data sharing, geological blockchain, attribute-based encryption, zero-knowledge proof, privacy protection

## Abstract

With the rapid development of geological blockchains and Internet of Things-based data acquisition technologies, massive amounts of heterogeneous data are constantly emerging. However, this data is stored in a distributed manner across different organizational or business blockchains. Data sharing among multiple geological blockchains faces numerous challenges, either exposing sensitive data during verification or lacking effective authorization mechanisms. Therefore, how to achieve fine-grained access control and privacy protection across multiple blockchains has become a critical issue that must be addressed in geological data sharing. In this paper, we propose GeoCross, a cross-chain geological data sharing framework that enables fine-grained authorization management and privacy protection. First, GeoCross provides a hierarchical hybrid encryption mechanism that uses symmetric encryption for geological data protection and ciphertext-policy attribute-based encryption to enable flexible cross-chain access policies. Second, we integrate a Groth16-based zero-knowledge proof mechanism, which allows a chain to verify the existence, integrity, and accessibility of off-chain data without revealing the content. Furthermore, we introduce a Reputation-based Non-interactive Relay node Selection protocol (RNRS), which enhances the trustworthiness and fairness of cross-chain routing. Finally, we implement GeoCross in a multi-chain Hyperledger Fabric environment and evaluate its performance under real-world workloads. Results show that Groth16 verification requires only three bilinear pairings, achieving a throughput of up to 390 tps on a single chain and 1550 tps in a concurrent multi-chain environment. Even with 50% malicious nodes, the RNRS protocol still maintains a success rate of over 91%. These results demonstrate that GeoCross provides an efficient and practical solution for secure and privacy-preserving cross-chain geological data sharing.

## 1. Introduction

Geological information, serving as fundamental data for natural resource surveys, energy exploration, and mineral development [1,2], directly influences industrial progress due to its authenticity and security. With the progress of geological information digitization, geological institutions and research institutes at various levels have accumulated massive data resources [3]. However, these data are distributed and stored in different bureaus or business systems, lacking effective collaboration mechanisms and serious data silos. Traditional cloud-based geological data sharing platforms rely on a centralized trust model, where a single service provider controls the storage, access management, and logging mechanisms. Even if cloud data is encrypted, this centralized architecture still creates a single point of failure and an attractive target for attacks or internal misuse. In addition, conventional authorization mechanisms typically grant users full data access once permission is issued, resulting in coarse-grained control and potential privacy exposure, especially in multi-organization geological data sharing.

Blockchain establishes a new trust foundation for geological data sharing [4]. Its decentralization and immutability enable multiple parties to achieve data sharing without relying on central authorities. Current research has attempted to apply blockchain to geological information management and research outcomes sharing scenarios, achieving data authorization and sharing through smart contracts [5]. However, the differences in trust models, access policies, and privacy-preserving mechanisms among different geological blockchain systems (such as scientific research chains, provincial bureau chains, and industry chains) pose challenges for data sharing among multiple blockchains.

Currently, some mainstream cross-chain schemes have been proposed [6,7,8,9], such as sidechain [10,11], notary mechanism [12,13], hash-locking [14,15], and relay chain [16]. These schemes achieve cross-chain collaboration by connecting independent blockchain networks, thereby alleviating the issues of single-chain performance and business interaction [17,18]. BitXHub [19], a cross-chain solution based on sidechain, supports asset transfer and information sharing across different blockchains. However, cross-chain data sharing requires exposing the data to all participants, which can lead to data privacy issues [20,21]. Wecross [22] proposed a simple and secure cross-chain scheme that adopts a gateway-like cross-chain architecture to enable interoperability. The gateway model still suffers from the drawbacks of centralized control [23], and the routing proxy node can access detailed information about cross-chain transactions, making its credibility a vulnerability that affects cross-chain security [24]. Therefore, most existing cross-chain solutions overlook the requirement for privacy protection and face the issue of sensitive data exposure [25,26], which may hinder business collaboration between multiple blockchains.

Since geological data involves national secrets, data privacy protection has become a key challenge in data sharing among multiple geological blockchains. On the one hand, the absence of a unified access control and authorization mechanism makes cross-chain data sharing subject to unauthorized access. On the other hand, the process of cross-chain data sharing may expose geological data and pose a risk of privacy leakage [27]. In addition, the proxy node that cross-chain communication relies on still faces issues such as trust opacity and unfair elections [28], making it difficult to ensure the security and reliability of data sharing.

In this work, to tackle the aforementioned issues, we propose GeoCross, a cross-chain geological data sharing scheme that enables fine-grained authorization and privacy protection. In contrast, GeoCross introduces zero-knowledge proofs (ZKP) based on Groth16, aiming to achieve privacy protection for cross-chain data. Furthermore, existing attribute-based encryption schemes are typically only applicable within a single governance domain and cannot directly handle cross-domain authorization needs between independent blockchains. Although some improved attribute-based encryption methods optimize performance, they still do not consider fine-grained authorization management for cross-chain access. GeoCross’s proposed hierarchical hybrid encryption mechanism decouples data encryption from authorization strategies, making it more suitable for cross-institutional and cross-chain data sharing environments. Moreover, existing cross-chain routing mechanisms generally employ fixed proxies or probabilistic selection strategies based on PoW/PoS, making them vulnerable to centralized, manipulated, or coordinated attacks. RNRS incorporates reputation values and verifiable random function (VRF) to achieve non-interactive agent election, improving the robustness and fairness of cross-chain routing. First, aiming at the complexities of cross-chain access authorization and the resulting privacy leakage, we propose a hierarchical access control mechanism for cross-chain geological data sharing. The hierarchical mechanism employs symmetric encryption (SE) at the data layer to secure data storage and sharing, and ciphertext-policy attribute-based encryption (CP-ABE) at the authorization layer to enable fine-grained management of cross-chain data access credentials, which effectively separates and coordinates data privacy protection and access policy control. Second, to balance trustworthiness and privacy protection during cross-chain data sharing, this paper proposes a cross-chain data verifiable mechanism based on ZKP. By constructing the Groth16 proof circuit [29], the owner chain can prove the existence, correctness, and accessibility of the data to the requester chain without revealing the plaintext data, thereby verifying the legitimacy of data before cross-chain interaction. This mechanism achieves “verifiable but invisible” privacy protection in cross-chain sharing, reducing the risk of privacy leakage and unnecessary communication overhead. Finally, to address the trust issues in cross-chain data sharing, this paper proposes RNRS. By combining node reputation with a random timer generation strategy, this mechanism autonomously elects proxy nodes without requiring interaction, thereby enhancing the trustworthiness of cross-chain relays. GeoCross facilitates secure and trustworthy data sharing between geological blockchains, fulfills the requirements for cross-chain collaboration with multi-source geological information. The main contributions of this paper are as follows:We propose a hierarchical hybrid encryption mechanism for cross-chain geological data sharing. At the data layer, SE is employed to enhance the efficiency of encrypting large-scale geological datasets, while at the authorization layer, CP-ABE is integrated to achieve fine-grained access control. By decoupling encryption from authorization, the mechanism enables fine-grained authorization and minimizes data exposure.We design a cross-chain data-verifiable and privacy-preserving scheme based on ZKP, which employs a Groth16 proof circuit to enable the requester chain to verify the existence, correctness, and availability of owner chain data during cross-chain interactions without accessing any data.We introduce a random RNRS based on reputation value and VRF. This mechanism utilizes dynamic updates of reputation and a VRF to achieve random and reliable agent election, improving the reliability of cross-chain routing.We conduct a security analysis focusing on data confidentiality and authorization correctness, and evaluate its performance in a multi-chain environment. The experimental results demonstrate the feasibility of GeoCross in geological data sharing scenarios.

The rest of this paper is organized as follows. Section 2 reviews related work. In Section 3, we present the system model, security model, and design goals. Section 4 presents the details of GeoCross. Section 5 and Section 6 discuss security analysis and performance evaluation, respectively. Finally, in Section 7, we conclude this paper.

## 2. Related Work

In recent years, blockchain technology has been gradually introduced into the field of geological data sharing due to its decentralization, immutability and traceability, to solve problems such as data silos, tampering risks and privacy leakage in centralized geological data management. Li et al. [30] investigated the application potential of blockchain in geological data sharing and believed that blockchain can effectively improve the credibility and tamper-resistance of geological data during storage and sharing processes. In response to the common problems of “duplicate collection” and “information silos” in current geological data sharing, Li et al. [31] proposed a sharing model that integrates geological data indexing and blockchain, putting an index on the chain to achieve secure sharing and traceable management. Zhou et al. [32] built a geological data sharing platform based on blockchain. By writing verified geological information and its ciphertext index into the blockchain and introducing a random competitive allocation mechanism, they ensured that the shared data could not be reverse-parsed under the premise of verifiability, thereby improving the security and fairness of the data sharing process.

Existing research indicates that blockchain has significant advantages in improving the security, transparency, and credibility of geological data sharing. Existing studies demonstrate that blockchain provides significant advantages in enhancing the security, transparency, and trustworthiness of geological data sharing. Razzaq et al. [33] proposed a decentralized blockchain-based architecture for managing remote-sensing big-data documents, supporting secure sharing, version control, and integrity protection without relying on a central authority. However, the framework is designed primarily for document-level remote-sensing data and focuses on immutability and availability, confidentiality is handled only via conventional encryption, and access policies cannot be defined or enforced in a fine-grained manner. Zhang et al. [34] developed a blockchain-based management scheme for Earth-observation products to improve decentralization, traceability, and tamper resistance of provenance metadata. Nevertheless, this solution is mainly concerned with provenance management rather than data privacy, and it does not address fine-grained authorization or cross-chain access, particularly in heterogeneous environments where different organizations operate independent blockchains and cloud platforms.

From a broader geospatial perspective, Chafiq et al. [35] conducted a comprehensive survey on blockchain applications in geospatial data sharing, highlighting its potential for secure storage, property rights management, and traceability, as well as key challenges such as scalability, privacy, interoperability, and regulatory constraints; however, most of these efforts remain conceptual. In more specific geological use cases, Aufaristama [36] explored the use of non-fungible tokens (NFTs) on public blockchains for geological data dissemination and digital asset management. Nevertheless, this work emphasizes openness and public dissemination rather than controlled sharing of sensitive geological information.

However, blockchain still has shortcomings in the field of geological data sharing [37,38]. First, most existing work focuses on data tamper-resistance and traceability, while paying insufficient attention to data confidentiality and access control mechanisms. Second, geological data often spans multiple departments and systems, and cross-institutional trust and privacy protection mechanisms remain not sound. The consensus efficiency and access permission of consortium blockchain under multi-node collaboration still exist. Furthermore, current research primarily focuses on on-chain data storage or indexing on chain, while further research is needed on the collaboration between encryption technologies and original data management methods such as blockchain and geological clouds.

## 3. System Overview

### 3.1. System Model

The architecture of GeoCross is illustrated in Figure 1, which includes four entities: the Owner Chain (Geoscience Institute Chain), the Requester Chain (Provincial Bureau Chain), the Relay Chain, and Geological Cloud Server (GCS). Furthermore, to enable cross-chain data sharing, each chain incorporates proxy nodes.

Owner Chain: Owner Chain consists of data owners, represented by the Geoscience Institute Chain that possess original data. It is responsible for responding to data requests and managing shared processing. Data owners encrypt the original data and issue data tokens to authorize cross-chain data sharing.Requester Chain: Requester Chain is a consortium chain (e.g., Provincial Bureau Chain) composed of data requesters. Data requesters seek to obtain and utilize correct data. They initiate access requests, verify data validity, request access permissions across the blockchain, and ultimately acquire the data ciphertext.Relay Chain: Relay Chain serves as the trust and collaboration hub of the cross-chain system. It is responsible for business chain registration, cross-chain message routing, and access policy verification, ensuring the credibility and privacy of cross-chain interoperability.Geological Cloud Server (GCS): GCS serves as a trusted storage service shared by all business chains, storing only ciphertexts and not any plaintext information [39].Proxy Nodes: Acting as cross-chain interfaces, proxy nodes are responsible for routing and forwarding cross-chain requests, data tokens, and ciphertexts, but do not perform any encryption operations or plaintext processing.

### 3.2. Design Goals

To address the Data Privacy Exposure (DPE) problem in cross-chain geological data sharing, this paper proposes GeoCross, a privacy-preserving cross-chain sharing scheme for geological blockchains that ensures secure data sharing, privacy protection, and verifiable access. Specifically:Cross-chain trust: To overcome the lack of inter-chain trust, GeoCross employs a Relay Chain and proxy node collaboration mechanism to implement policy verification and data authorization across multiple blockchains. The Relay Chain supports business chain registration, message routing, ensuring that cross-chain requests are legitimate and secure interactions. Proxy nodes within each Owner Chain and Requester Chain are selected randomly and with minimal trust using verifiable algorithms. They only handle message forwarding and token verification without accessing any plaintext data, achieving a minimal-trust proxy model.Cross-chain privacy protection: To mitigate Cross-chain Privacy Exposure (CPE), GeoCross integrates a zero-knowledge verification mechanism, enabling the Owner Chain to prove the existence and accessibility of geological data to the Requester Chain without revealing any original data.Cross-domain access control: To prevent Cross-domain Data Unauthorized Access (CDUA), the system adopts a multi-layered authorization encryption strategy, ensuring that only users whose attributes satisfy policies can perform hierarchical decryption. This design guarantees clear authority boundaries and controlled access throughout the cross-chain data sharing.

### 3.3. Threat Model

We assume that the central authority and attribute authorities are fully trustworthy [40,41], they are capable of correctly executing the system initialization algorithm, securely generating global system parameters and the master key, and distributing attribute private keys to authenticated users exclusively through secure channels. The data owner is assumed to be honest and authorizes data transmission exclusively to authorized users, but all on-chain communications are subject to potential monitoring. Data storage modules are honest-but-curious, they faithfully store data and respond to the requests from both the Owner Chain and the Requester Chain, but they are also curious about the stored data.

Potential attackers in the system include malicious data users, malicious proxy nodes, and the attacked Requester Chain. Malicious data users fall into two types: (1) unauthorized users, who attempt to bypass access control policies to directly access data ciphertext across chains; (2) authorized users, who may abuse their own attribute private keys or collude with proxy nodes to obtain data beyond their permissions. They attempt to obtain unauthorized data by cracking the symmetric key $K_{data}$.

Malicious behaviors of proxy nodes include refusing to forward services, tampering or forging data tokens, and attempting to recover plaintext content from forwarded data tokens and ciphertext.

The Relay Chain or other business chains are at risk of being attacked. They may forge verification results for access control or incorrectly transmit data ownership proofs, thereby affecting the system’s cross-chain data sharing services.

To further analyze the system’s resistance to attacks, we define two types of typical adversaries:Type I (external attacker): Without a legitimate attribute key, they can only attempt to obtain tokens or data by cracking ciphertext, forging requests, or eavesdropping on communications.Type II (internal attackers): Include malicious data users, malicious proxy nodes, and compromised business chains. They may attempt to hinder data sharing and compromise data privacy by disrupting cryptographic algorithms or communication transmission.

## 4. Detail of Our Proposed Scheme

### 4.1. System Initialization

Trusted Setup Phase: In this phase, the authority (e.g., the China Geological Survey) generates the proving key prk and verification key vk for geological blockchains (both data owners and data users), which are related to the verification of geological data. The vk produced in this phase contains the bilinear group parameters $\left(G_{1},G_{2},G_{3}\right)$ and essential secret parameters α,β,γ,δ. All these parameters are stored within the Relay Chain.

Applying the ABE scheme in inter-blockchain data sharing requires a Key Generation Center (KGC) to initialize system parameters. The KGC serves as a trusted third-party authority responsible for executing the initialization algorithm that generates the cryptographic parameters required.(1)$$\begin{array}{ccc} PP & \leftarrow & ABE.Setup(1^{\lambda })\\ SK & \leftarrow & ABE.MSKGen(PP) \end{array}$$

The KGC employs a hash function *H* to compute the public parameters. The function *H* maps arbitrary binary strings to elements of group *G*:*H*:${\left\{0,1\right\}}^{*}\to G$. In GeoCross, *H* takes two types of inputs: (x,l,t), where *x* is an arbitrary string; (j,l,t), where *j* is a positive integer. Here, l∈{1,2,3} and t∈{1,2}. For brevity, these inputs are denoted as xlt and 0jlt, respectively. The detailed process and resulting outputs of the KGC are as follows:

$GlobalSetup(1^{\lambda })\to PP$: The authority creates a bilinear pairing tuple $\left(e,G,G_{T},g,p\right)$ and selects a hash function $H_{1}$:$G_{T}\to {\left\{0,1\right\}}^{*}$ involved in the encryption process to compute the public parameters, which maps any binary string to an element in group *G*. In this scheme, *H* has two types of inputs, namely (x,l,t) and (j,l,t), where *x* is an arbitrary string, j is any positive integer, l∈{1,2,3}, and t∈{1,2}. We denote these two inputs as xlt and 0jlt. In the security analysis, these hash functions are regarded as random functions. Furthermore, by randomly selecting g∈G and h∈H, the global public parameters are obtained as $pp=(p,G,H,G_{T},e,g,h,H_{1})$.

$AuthSetup\left(1^{\lambda }\right)\to \left(pk,msk\right)$: In this process, the initialization function randomly selects parameters $a_{1}$ and $a_{2},b_{1}$ and $b_{2}$ from a cyclic group of prime order *p*, and parameters $d_{1},d_{2},d_{3}$ from $Z_{p}$. Then, *Setup* uses the global parameters to generate the public key $pk:=(h,\;H_{1}:=h^{a_{1}},\;H_{2}:=h^{a_{2}},\;T_{1}:=e{\left(g,h\right)}^{d_{1}a_{1}+d_{3}},\;T_{2}:=e{\left(g,h\right)}^{d_{2}a_{2}+d_{3}})$ and master private key $msk:=(g,\;h,\;a_{1},\;a_{2},\;b_{1},\;b_{2},\;g^{d_{1}},\;g^{d_{2}},\;g^{d_{3}})$ for the ABE scheme [42].

$KeyGen\left(msk,S\right)\to sk:=(sk_{0},{\left\{sk_{y}\right\}}_{y\in A},sk^{\prime })$. For each data user with attribute set A∈S, the authority uses the *KeyGen* algorithm to generate the attribute-based private key sk, taking the master private key msk and the node’s attribute set *S* as input. The KGC first randomly selects parameters $r_{1}$ and $r_{2}$ from $Z_{p}$, and then computes the partial private key $sk_{0}:=\left(h^{b_{1}r_{1}},\;h^{b_{2}r_{2}},\;h^{r_{1}+r_{2}}\right)$.

Then, for each y∈S and t=1,2, the authority uses the corresponding parameters $h,b_{1},b_{2}$ from msk to compute the partial private keys $sk_{\left(y,t\right)}$ for different attributes.(2)$$\begin{array}{c} {\mathrm{sk}}_{y,t}:=H{\left(y1t\right)}^{\frac{b_{1}r_{1}}{a_{t}}}\cdot H{\left(y2t\right)}^{\frac{b_{2}r_{2}}{a_{t}}}\cdot H{\left(y3t\right)}^{\frac{r_{1}+r_{2}}{a_{t}}}\cdot g^{\frac{\sigma_{y}}{a_{t}}} \end{array}$$

Then, compute the partial private key $sk_{y}:=\left(sk_{y,1},\;sk_{y,2},\;g^{-\sigma_{y}}\right)$, where a random value $\sigma_{y}$ is selected from $Z_{p}$. And authority computes $sk_{t}^{\prime }:=g^{d_{t}}\cdot H{\left(011t\right)}^{\frac{b_{1}r_{1}}{a_{t}}}\cdot H{\left(012t\right)}^{\frac{b_{2}r_{2}}{a_{t}}}\cdot H{\left(013t\right)}^{\frac{r_{1}+r_{2}}{a_{t}}}\cdot g^{\frac{\sigma^{\prime }}{a_{t}}}$, for t=1,2, $\sigma^{\prime }\overset{R}{\leftarrow }Z_{p}$, ${\mathrm{sk}}^{\prime }=\left({\mathrm{sk}}_{1}^{\prime },\;{\mathrm{sk}}_{2}^{\prime },\;g^{d_{3}},\;g^{-\sigma^{\prime }}\right)$. Finally, the secret key of data user is $\left(sk_{0},\;{\left\{sk_{y}\right\}}_{y\in A},\;sk^{\prime }\right)$.

### 4.2. Proxy Node Selection

To enable cross-chain data access, data sharing between geological blockchains and the Relay Chain is carried out through proxy nodes. The proxy’s public key is registered on the Relay Chain via a signed blockchain transaction. Since all chains obtain this public key directly from the immutable ledger rather than through off-chain message exchange, the process is inherently resistant to man-in-the-middle attacks. We propose RNRS to elect proxy nodes for each geological blockchain. All nodes are eligible to participate in the election. RNRS divides the election process into multiple epochs, each containing a fixed number δ blocks. A portion of these blocks (δ×cl to δ, 0≤cl≤1) is reserved for the election phase. Each candidate node combines its reputation value with the latest checkpoint block hash to generate a random number using a VRF, which determines its waiting timer length *L*. The node whose timer expires first becomes the proxy node for this epoch, broadcasts the election result, and other nodes stop their timers.

In RNRS, each node’s reputation value is maintained and updated uniformly on the blockchain. Initially, all nodes are assigned the same reputation value $r_{i}$, a node’s reputation increases by 1 point each time it successfully participates in an intra-chain consensus. If a node is absent during consensus, 2 points are deducted. When a node’s reputation value drops to zero, it loses its eligibility to participate in subsequent elections. To prevent any single node from the proxy for extended periods, the reputation value of a node is immediately halved upon election, thereby increasing the likelihood that other nodes will be selected in future rounds.

Each candidate node *i* constructs a timer by first obtaining its own $r_{i}$ and the total reputation value of all nodes *R*. Using the latest checkpoint block $c_{e}$ in the election as a random seed, the node generates a VRF random number and its proof $\left(y,\pi_{y}\right)=GenVRF\left(c_{e},sk_{i}\right)$, where $sk_{i}$ is the private key of node *i* and the corresponding public key $pk_{i}$ is registered on the blockchain.

Each candidate proxy node then uses the function $Hash\left(y,L\right)<d\times \frac{r_{i}}{R}$ to find the smallest timer value *L* that satisfies the condition, where *d* is a difficulty parameter. Nodes with higher $r_{i}$ have a wider range of valid counters, making them more likely to be selected as proxy nodes. Assuming the first block height of the election period is $H_{e}$, where *i* is elected when the block at height $H_{e}+L$ is added to the ledger. Upon receiving node *i*’s election, upper-layer nodes verify the following conditions before confirming the election result: (a) whether $Hash\left(y,L\right)<d\times \frac{r_{i}}{R}$ satisfies the inequality above; (b) whether the current block height has exceeded $H_{e}+L$; (c) whether the proof $\pi_{y}$ correctly verifies that *y* was generated from the random seed $c_{e}$. If checks pass and no other node has already been elected, node *i* is designated as the proxy node for this epoch, responsible for cross-chain interoperability until a new representative is elected or the next epoch begins. In exceptional cases where all candidates produce excessively large values *L* and no suitable proxy can be selected, the election is considered timed out, and the node with the highest reputation value is assigned as the proxy node.

The reputation mechanism ensures fairness in proxy selection and gradually eliminates malicious or inactive nodes. This approach provides a reliable transmission process for cross-chain data sharing among geological blockchains. Subsequent cross-chain data exchanges are securely forwarded through the elected proxy nodes.

### 4.3. Cross-Chain Data Request

Phase 1: Cross-chain Request Initiation. When a provincial bureau user (i.e., data user) attempts to initiate a data access to another geological blockchain on their chain (Provincial Bureau Chain), the access request is processed by invoking a cross-chain smart contract. The user submits its desired data identifier ($D_{id}$) as the target parameter of the request. Upon detecting the cross-chain request, the provincial chain’s proxy node triggers the corresponding cross-chain data access. Specifically, the proxy node first collects and verifies the user’s attributes. These attributes include the user’s identity information, role permissions, and so on. The proxy node then signs the access request, which consists of the request identifier ($Request_{ID}$), the data identifier ($D_{id}$), and the user’s attributes with the Provincial Bureau Chain’s private key. Finally, the proxy node forwards the request to the Relay Chain through the routing interface configured by the Relay Chain.

Phase 2: Data Existence Verification. When the access contract on the Relay Chain receives a cross-chain request from the Provincial Bureau Chain, it first performs integrity and validity verification of the request message. The Relay Chain’s nodes use the public key of the Provincial Bureau Chain to verify the digital signature submitted by the proxy node, ensuring that the message has not been tampered with and it was indeed initiated by a legitimate provincial-chain entity. If the signature verification succeeds, the access contract further reviews the request based on the fine-grained access control policy. The contract compares the requester’s user attributes with the policy associated with the target data $D_{id}$. If satisfied, the Relay Chain routes the access request to the target proxy node of the Geological Institute Chain, which serves as the entry point for cross-chain communication. Otherwise, if the access policy is not satisfied, the request is rejected.

Then, the proxy node of the Geoscience Institute Chain queries its local state to confirm that $D_{id}$ exists and is accessible. To further prove the existence and accessibility of the geological data $D_{id}$ to the Provincial Bureau Chain, a ZKP $Z_{\pi }$ must be generated. The Geoscience Institute Chain’s proxy node collects the private witness (including the Merkle path path, status value status = 1, etc.) and runs the Groth16 algorithm to prove possession and accessibility without revealing any geological information. The prover first converts the data verification circuit into a Quadratic Arithmetic Program (QAP), then uses prk together with the private inputs to generate the proof $Z_{\pi }=\left(A,B,C\right)$, where $A\in G_{1}$ is the commitment to path, $B\in G_{2}$ is the commitment to status, and $C\in G_{1}$ contains the cross-checking for all constraints.

To compute and verify the ZKP, the scheme designs a core constraint logic circuit. The circuit takes the current public Merkle root root_hash of the Geoscience Institute Chain and the $D_{id}$ as public inputs; private inputs include the leaf hash of $D_{id}$, leaf_hash = SHA-256 ($D_{id}$, metadata_hash), the Merkle path indices path_index[n], sibling hashes sibling_hashes[n], and the data accessibility status. Circuit constraints and computation flow:Leaf verification: compute calculated_leaf_hash = SHA-256$\left(D_{id},metadata\_hash\right)$ and constrain calculated_leaf_hash = leaf_hash to ensure consistency with the leaf node.Merkle path verification: starting from leaf_hash and sibling_hashes[0], recursively compute each level according to path_index[i] until obtaining calculated_root, and constrain calculated_root = root_hash to prove that $D_{id}$ belongs to the chain’s Merkle tree.Accessibility verification: constrain status = 1 to confirm the data is accessible.

When all constraints hold, the circuit outputs 1, indicating that the verification passes and the proof is valid.

Then, the Geoscience Institute Chain routes the generated proof tuple $Z_{\pi }=\left(A,B,C\right)$ to the Provincial Bureau Chain via the Relay Chain. In this process, the Relay Chain only performs message forwarding, ensuring the privacy of geological data. The proxy node of the Provincial Bureau Chain uses vk issued by the Geoscience Institute Chain to verify the validity of the proof $Z_{\pi }$.

After receiving the proof, the proxy of the Provincial Bureau Chain first extracts the target data identifier $D_{id}$ and the published Merkle root root_hash from the request, along with other public inputs provided by the Geoscience Institute Chain.

It uses the verification parameters $\left\{w_{i}\right\}$ contained in vk, the verifier encodes these public inputs into a group element $W=\prod w_{i}^{\left(x_{i}\right)}\in G_{1}$, where each $x_{i}$ corresponds to a specific public input value. Finally, the verifier uses the verification function to check the proof correctness: $isValid=VerifyProof(vk,Z_{\pi },\left[root\_hash,D_{id}\right])$.

If isValid=true, the Provincial Bureau Chain confirms that the geological data $D_{id}$ exists and is accessible on the Geoscience Institute Chain, without learning any private information.

The proxy node of the Provincial Bureau Chain verifies the proof $Z_{\pi }$ with vk by performing three bilinear pairing computations. First, it computes e(A,B), then separately evaluates $e(g^{\alpha },h^{\beta })$, $e(g^{\gamma },h^{\delta })$, and $e(C,h^{z})$. Only $Z_{\pi }$ generated by the legitimate data owner satisfy $e\left(A,B\right)=e{\left(g,h\right)}^{\left(\alpha \beta +\gamma \delta \right)}\cdot e\left(C,h^{z}\right)$, which confirms that the prover indeed possesses the claimed data. If the verification succeeds, the Provincial Bureau Chain is assured that the Geoscience Institute Chain truly holds the geological data and the data status is valid. If the verification fails, the Provincial Bureau Chain rejects the cross-chain request.

Phase 3: Owner-side Encryption and Token Issuance. After confirming that the requested data exists and is accessible, the data owner DO must ensure data confidentiality and fine-grained access control in cross-chain sharing. GeoCross employs a two-stage encryption mechanism combined with a defined access control policy to generate the corresponding access token.

First, the DO encrypts the original geological data using a symmetric key $K_{data}$ to produce the ciphertext CipherText, which is stored in a GCS. Then, to protect the $K_{data}$, the DO encrypts it using an ABE scheme. Using pk, the access policy matrix *M*, the DO computes:(3)$$\begin{array}{c} Enc\left(K_{data}\right)=ABE.Enc\left(pk,K_{data},\left(M,\pi \right)\right) \end{array}$$

This process ensures that only users whose attributes satisfy the access policy can decrypt and use the $K_{data}$.

The specific steps are:Step 1: Randomly select parameters $s_{1},s_{2}$ from $Z_{p}$ and compute the corresponding values used for key encryption.Step 2: Given an attribute matrix *M* with i=1,…,5 and l∈1,2,3, compute $ct_{i,\ell }=H{\left(\pi \left(i\right)\ell 1\right)}^{s_{1}}\cdot H{\left(\pi \left(i\right)\ell 2\right)}^{s_{2}}\cdot \prod_{j=1}^{3}{\left[H{\left(0j\ell 1\right)}^{s_{1}}\cdot H{\left(0j\ell 2\right)}^{s_{2}}\right]}^{{\left(M\right)}_{i,j}}$ that can be derived for each attribute dimension.Step 3: Finally, compute $ct^{\prime }$, which represents the encryption of $K_{data}$. Thus, the cross-chain ciphertext can be expressed as: $CT=(ct_{0},ct_{1},\ldots ,ct_{n},ct^{\prime })$.

This ciphertext CT is embedded in the cross-chain message Msg and forwarded through the Relay Chain, ensuring data confidentiality and fine-grained access in the GeoCross.

After two stage encryption, the system designs a Data Ticket mechanism to enable controlled authorization. The Owner Chain issues a DataTicket = {Request_ID, $D_{id}$, Expiry, Location(CipherText)}, where Request_ID represents the identifier of the cross-chain request, $D_{id}$ represents the identifier of the target data, Expiry defines the token’s validity period, and Location(CipherText) represents the ciphertext location stored in the GCS.

### 4.4. Cross-Chain Data Access

Once the Requester Chain possesses a valid, unexpired data token (Data Ticket), it can retrieve the corresponding ciphertext from GCS based on the storage location provided in the token. Furthermore, the proxy node also retrieves $Enc(K_{data})$ from the Relay Chain. After data retrieval, the requesting proxy node packages the request result {CipherText, $Enc(K_{data})$, Ticket, signature verification record} and writes it to the Requester Chain for on-chain recording. The actual *DU* attempt to decrypt $Enc_{SK_{attr}}\left(K_{data}\right)$ with the $SK_{attr}$ corresponding to its own attributes. Only if its attributes satisfy the access control conditions can the *DU* recover the key and decrypt the original data; otherwise, the request is terminated.(4)$$\begin{array}{c} \mathrm{ABE}.\mathrm{Dec}\left(pk,CT,sk\right)\to K_{data} \end{array}$$

Step 1: When the attribute sets accessing the blockchain satisfy the attribute matrix (M,π) used for encryption, there will be a constant $\gamma_{i}$.

Step 2: According to the constant, we can compute:(5)$$\begin{array}{cc} \mathrm{num} & :=ct^{\prime }\cdot e\;\left(\prod_{i\in I}ct_{i,1}^{\gamma_{i}},\;{\mathrm{sk}}_{0,1}\right)\cdot e\;\left(\prod_{i\in I}ct_{i,2}^{\gamma_{i}},\;{\mathrm{sk}}_{0,2}\right)\cdot e\;\left(\prod_{i\in I}ct_{i,3}^{\gamma_{i}},\;{\mathrm{sk}}_{0,3}\right),\\ \mathrm{den} & :=e\;\left({\mathrm{sk}}_{1}^{\prime }\cdot \prod_{i\in I}{\mathrm{sk}}_{\pi (i),1}^{\gamma_{i}},\;ct_{0,1}\right)\cdot e\;\left({\mathrm{sk}}_{2}^{\prime }\cdot \prod_{i\in I}{\mathrm{sk}}_{\pi (i),2}^{\gamma_{i}},\;ct_{0,2}\right)\cdot e\;\left({\mathrm{sk}}_{3}^{\prime }\cdot \prod_{i\in I}{\mathrm{sk}}_{\pi (i),3}^{\gamma_{i}},\;ct_{0,3}\right). \end{array}$$

The cross-chain data Msg=num/den can be decrypted. Finally, DU can use $K_{data}$ to decrypt the original geological data. The process of cross-chain data sharing is shown in Figure 2.

## 5. Security Analysis

### 5.1. Data Confidentiality

**Theorem** **1.**
*Data Confidentiality: GeoCross ensures data confidentiality and authorized access through a hybrid encryption policy combining symmetric encryption (SE) and attribute-based encryption (ABE). Specifically, the data owner encrypts the plaintext M using a randomly generated symmetric key $K_{data}$ to produce $C_{sym}$ = $SE.Enc_{K_{data}}\left(M\right)$. Then, $K_{data}$ is encrypted with an access policy Policy using ABE, get $C_{ABE}$ = $ABE.Enc(K_{data},Policy)$. The ciphertext is expressed as $(C_{sym}$,$C_{ABE}$), where $C_{sym}$ is maintained off-chain, and $C_{ABE}$ is transferred across blockchains through the Relay Chain.*


**Proof.** To demonstrate that no probabilistic polynomial-time (PPT) adversary can compromise data confidentiality with a non-negligible advantage, we construct a security game $Game_{Conf}$ between the adversary A and the challenger C. This game models the confidentiality guarantees of GeoCross under adaptive attacks. Assume that A achieves a non-negligible advantage $Adv_{Game_{Conf}}$ in this game. The adversary selects a challenge matrix with each dimension bounded by *q*. A is allowed to query any secret key that cannot be directly employed to decrypt the challenge ciphertext generated by C. The detailed procedure is described as follows.

Setup: C executes *ABE.Setup* to generate the public parameters PP and publishes the system configuration.Query Phase I: Attribute Key Queries. A may adaptively request the private key associated with any attribute set $S_{i}$. If $S_{i}⊈Policy$, C provides the corresponding key $SK_{\left(S_{i}\right)}$.Challenge Phase: A submits a pair of equal-length plaintexts ($M_{0}$,$M_{1}$). C selects a random bit b∈{0,1}, generates a random symmetric key $K_{data}$, and computes: $C_{sym}=SE.Enc_{\left(K_{data}\right)}\left(M_{b}\right)$, $C_{ABE}=ABE.Enc\left(K_{data},Policy\right)$. It returns the ciphertext pair $\left(C_{ABE},C_{sym}\right)$ to A.Query Phase II: A may continue to request private keys for attribute sets that do not satisfy Policy, and C responds as before.Guess Phase: Finally, A outputs a guess $b^{\prime }$. The adversary’s advantage is defined as: $Adv_{Game_{Conf}}\left(A\right)=\left|Pr\left[b^{\prime }=b\right]-\frac{1}{2}\right|$.

The scheme is IND-CPA secure under the chosen-policy model if $Adv_{Game_{Conf}}\left(A\right)$ is negligible for all *PPT* adversaries A.

Hybrid Experiments

To verify the security of the proposed scheme, we employ a hybrid experiment to reduce the security of the algorithm.
$H_{0}$: Real Game. The hybrid $H_{0}$ corresponds to the real security game $Game_{Conf}$, which accurately models the confidentiality properties of GeoCross under the defined threat model.$H_{1}$: In $H_{1}$, the symmetric encryption operation is replaced by an ideal random oracle. If A can distinguish $H_{0}$ and $H_{1}$ with non-negligible probability, it can be used to break the *IND-CPA* security of the ABE scheme under the chosen-policy attack. Therefore, $|Pr\left[A\;\mathrm{wins}\;\mathrm{in}\;H_{0}\right]-Pr\left[A\;\mathrm{wins}\;\mathrm{in}\;H_{1}\right]|\leq Adv_{SE}^{IND-CCA}\left(\lambda \right)$.$H_{2}$: In $H_{2}$, the ABE process is idealized. Instead of encrypting the symmetric key $K_{data}$, the challenger encrypts a uniformly random value $U\leftarrow {\left\{0,1\right\}}^{|K_{data}|}$ under a random policy Policy: $C_{ABE}=ABE.Enc\left(U,Policy\right)$. If the adversary can distinguish $H_{1}$ and $H_{2}$, an adversary can be built to break the security of the ABE scheme under the CPA model, which contradicts the security assumption.$H_{3}$: In $H_{3}$, the ciphertext $\left(C_{ABE},C_{sym}\right)$ becomes statistically independent of the challenge plaintext $M_{b}$. Finally, the adversary’s distinguishing advantage is zero. Therefore, the advantage of any PPT adversary A against the GeoCross confidentiality model satisfies: $Adv_{Game_{Conf}}\left(A\right)\leq Adv_{SE}^{IND-CCA}+Adv_{ABE}^{IND-CPA}=negl\left(\lambda \right)$, indicating that GeoCross achieves semantic security under standard cryptographic assumptions.□

**Lemma** **1.**
*In the GeoCross, any entity that fails to satisfy the predefined access policy, including passive eavesdroppers, off-chain storage servers (GCS), relay/proxy nodes, and unauthorized users, are unable to recover the $K_{data}$ or M, even with full access to both the off-chain ciphertext and cross-chain metadata.*


Collusion-Resistance: If an attribute set does not satisfy the access policy Policy, the ABE scheme satisfying selective-policy IND-CPA security ensures that $ABE.Enc(K_{data},Policy)$ is indistinguishable to unauthorized entities, making $K_{data}$ pseudorandom to them. Furthermore, collusion among multiple users’ attribute private keys still cannot satisfy the access control policy, and thus $K_{data}$ cannot be recovered. Therefore, unauthorized users or colluding parties cannot compromise the ABE scheme.

Honest-but-Curious GCS/Relay/Proxy: Although the *GCS*, Relay Chain, and proxy nodes participate in the cross-chain data sharing, they can only observe ($C_{sym}$,$C_{ABE}$) and the metadata. The proxy node merely forwards $\left(Z_{\pi },C_{ABE},C_{sym}\right)$ or authorized content generated through the relay process, while the GCS only stores encrypted data. They cannot recover $K_{data}$ or *M*. From the hybrid experiment, it follows that for any adversary A, the following holds: $|Pr\left(A\left(M_{0},M_{1}\right)=b\right)-1/2|\leq Adv_{SE}^{IND-CCA}+Adv_{ABE}^{IND-CPA}=negl\left(\lambda \right)$. Therefore, cross-chain data confidentiality is guaranteed under the unauthorized adversary model.

### 5.2. Data Correctness

**Theorem** **2.**
*Cross-chain Data Correctness: To prove that owner chains (e.g., geological institute chains) can access data without revealing any content, GeoCross uses a ZKP-based method. In this section, we verify the completeness, knowledge soundness, and zero-knowledge properties of this method. The requesting chain can determine whether other chains possess accessible data solely from the tuple $\left(vk,stmt,Z_{\pi }\right)$, while learning nothing about any private information contained in $Z_{\pi }$. Prove that the cross-chain data is verifiable and not exposed, which means the requester chain can determine whether owner chains possess accessible data based on $\left(vk,stmt,Z_{\pi }\right)$, without obtaining any private information from the proof $Z_{\pi }$.*


Assume (prk,vk) are the trusted parameters of the zero-knowledge proof system. Let R(stmt,w) be the zero-knowledge defined in GeoCross, where the public input is $stmt=(root_{hash},D_{id})$ and the private witness is *w* = ($leaf_{hash}$, $metadata_{hash}$, $path_{index}$, $sibling_{hashes}$, status). The relation holds if the following conditions are satisfied: (1) compute $leaf_{hash}$ from $\left(D_{id},metadata_{hash}\right)$; (2) using $path_{index}$ and $sibling_{hashes}$, reconstruct bottom-up the Merkle root calculated_root and ensure $calculated\_root=root_{hash}$; (3) “status” = 1. If and only if (stmt,w)∈R, the Groth16 proof $Z_{\pi }=\left(A,B,C\right)$ could be verified, i.e., $Verify(vk,stmt,Z_{\pi })=1$; and it satisfies: completeness, knowledge soundness, and zero-knowledge.

*Completeness*: The data owner acts as the prover, possessing data access rights, and can successfully respond to verification requests initiated by a requester. To verify the completeness, we define a security game $Game^{COMP}$: the challenger C generates $\left(prk,vk\right)\leftarrow Setup\left(1^{\lambda },R\right)$, and the prover P holds a relation (stmt,w)∈R. If P outputs a proof $Z_{\pi }\leftarrow Prove\left(prk,stmt,w\right)$ such that $Verify(vk,stmt,Z_{\pi })=1$, then P wins the game, it possesses legitimate data access rights.

**Proof.** When the QAP constraints are satisfied, there exists a polynomial t(x) such that $e\left(A,B\right)=e{\left(C,h\right)}^{t}\times F\left(vk,stmt\right)$, where F (vk,stmt) denotes the combination term of public input polynomials. If the GeoCross circuit correctly compiles the Merkle hash constraints and R(stmt,w)=1, then t(x) exists and the equality $e\left(A,B\right)=e{\left(C,h\right)}^{t}\times F\left(vk,stmt\right)$ holds. Thus, the probability that the verification succeeds is: $Pr(Verify\left(vk,stmt,Z_{\pi }\right)=1)=1$. Therefore, the verification is passed, the scheme completeness is proved. □

Knowledge-Soundness. The prover (i.e., the Owner Chain) must hold a correct witness to convince the verifier (i.e., the Requester Chain) that it holds a valid Merkle path with status=1, which guarantees that the verifier accepts only when a valid witness w satisfying R(stmt,w)=1 exists. To verify the knowledge soundness, we define the security game $Game^{KS}$: A malicious forger A attempts to generate a fake proof $Z_{\pi }^{\prime }$ that convinces the verifier of the existence or accessibility of data with a invalid witness *w*. If A can compute $\left(stmt,Z_{\pi }\right)$ with vk such that $Verify(vk,stmt,Z_{\pi })=1$, while no witness *w* exists for which R(stmt,w)=1, the forger A wins the game.

**Proof.** Groth16 guarantees the existence of an efficient extractor E, whenever a malicious adversary A can generate a valid proof $Z_{\pi }$, E can extract the corresponding witness w. We construct a reduction algorithm B as follows: B simulates the Setup phase, generates (prk,vk), and sends vk to A. The adversary outputs a forged pair $\left(stmt,Z_{\pi }\right)$ such that $Verify(vk,stmt,Z_{\pi })=1$, implying that it has created an apparently valid proof. B uses the extractor E to derive a witness $w^{\prime }$. If $R(stmt,w^{\prime })=0$, then B has successfully broken the soundness, hence:$$\begin{array}{cc} Adv_{A}^{\mathrm{KS}}=\mathrm{Pr}[\mathrm{Verify}\left(vk,\mathrm{stmt},Z_{\pi }\right)\; & =1\wedge ∄w:R\left(\mathrm{stmt},w\right)=1]\leq {\mathrm{Adv}}^{\mathrm{Groth}16-\mathrm{KS}}+\mathrm{negl}\left(\lambda \right)\\ \; & =\mathrm{negl}(\lambda ). \end{array}$$This proof shows that if the verifier accepts $Z_{\pi }$, the system guarantees the existence of a Merkle proof path and a valid state. A malicious geological blockchain cannot forge proofs of false data existence or unauthorized access to deceive data requesters. □

Zero Knowledge. In GeoCross, the prover (e.g., a geological blockchain) can convince the verifier that a specific data exists and is in an accessible state without revealing any geological data. For any verifier *V*, there exists an efficient simulator S capable of generating a simulated proof $Z_{\pi }^{\prime }$ that is computationally indistinguishable from real proof $Z_{\pi }$, without accessing any private data. To verify the zero-knowledge, we define the security game $Game^{ZK}$: The challenger C generates (prk,vk) and selects a random bit *b*. If b=0, the proof is generated by $Z_{\pi }\leftarrow Prove\left(prk,stmt,w\right)$; if b=1, the simulator generates $Z_{\pi }^{\prime }\leftarrow Sim\left(vk,stmt\right)$. The adversarial verifier D is given $\left(stmt,Z_{\pi_{b}}\right)$ and outputs a guess $b^{\prime }$. The advantage of D is defined as: $Adv_{D}=\left|Pr\left[b^{\prime }=b\right]-\frac{1}{2}\right|$.

**Proof.** Only the values ($root_{hash}$,$D_{id}$) are publicly revealed, while all private inputs, such as the Merkle path and status are hidden. The proof $Z_{\pi }$ consists of the (A,B,C). For any external observer, these commitments are indistinguishable from those generated in a real execution. According to the zero-knowledge of Groth16, the probability can be expressed as $|Pr\left[D\left(H_{0}\right)=1\right]-Pr\left[D\left(H_{1}\right)=1\right]|\neq negl\left(\lambda \right)$. Therefore, the adversarial verifier’s distinguishing advantage is negligible, namely $Adv_{D}^{Z}K\leq negl\left(\lambda \right)$. □

The proof demonstrates that the prover (the Owner Chain) can convincingly prove to the verifier (the Requester Chain) the existence and accessibility of geological data without disclosing any original data, achieving data privacy protection in cross-chain sharing. Furthermore, ZKP verification is a cubic bilinear pairing operation that can be completed with low latency, providing privacy protection for geologically sensitive data and adapting to the verification-authorization workflow in high-privacy scenarios such as geological results and remote sensing data.

### 5.3. Security Analysis of RNRS

GeoCross uses RNRS to elect proxy nodes for cross-chain relay. To verify the security of the proposed reputation-based non-interactive random node selection method, this section evaluates node behavior and election results. The experimental environment consists of 20 nodes, one-third of which are malicious and the rest are honest. The initial reputation of all nodes is set to 100. Each experiment runs for 20 epochs and each consisting of 100 blocks. The election is triggered after consensus is reached on the 50th block. We compare the proposed RNRS with three other selection methods: random selection, round-robin election, and proof of wait stake-based selection.

As shown in Figure 3a, during the initial epochs, the election success rates of the various schemes varied little. As the number of epochs increased, the election success rate of this scheme increased rapidly, stabilizing at 95% after 20 epochs. This result demonstrates that RNRS can distinguish between honest and malicious nodes: honest nodes accumulate reputation through active participation in consensus, making them more likely to be elected as representatives in subsequent elections, while malicious nodes experience a decline in reputation due to refusal to participate or voting errors. This approach ensures the trust and secure selection of proxy nodes.

To further verify the security of RNRS, we simulated election results with a higher proportion of malicious nodes. As shown in Figure 3b, with the proportion of malicious nodes increased from 10% to 50%, the RNRS election success rate only dropped from approximately 96% to 91%. Its reputation accumulation, combined with the random function mechanism, effectively prevents malicious nodes from gaining an unfair advantage through computing power or frequent participation, maintaining the randomness and decentralization of this election method.

## 6. Performance Evaluation

In this section, we evaluate the performance of GeoCross from three perspectives: computational overhead, communication overhead, cross-chain interoperability latency, and compare it with existing approaches to demonstrate the feasibility of the proposed system.

### 6.1. Experimental Environment

For performance analysis, we selected three *q*-order groups, *G*, *H*, and $G_{t}$, whose bilinear pairings are defined as $e:G\times H\leftarrow G_{t}$. These groups are defined on the 256-bit Edwards curve MNT224, with the equation $y^{2}=x^{3}+1+d*x^{2}y^{2}$, which provides 96-bit security. Table 1 lists the computational time of various encryption operations on groups *G* and *H*. It can be seen that the cost of multiplication, exponentiation and hash computation on group *G* is significantly better than that on group *H*. Furthermore, GeoCross chooses AES-256 as the symmetric encryption algorithm and SHA-256 as the universal hash function. We deployed three blockchains (Hyperledger Fabric V2.2, Docker containers executing chaincode, and Raft as the consensus algorithm): two functional blockchains represent the Requester Chain (Provincial Bureau Chain) and Owner Chain (Geoscience Institute Chain), which simulate participants to implement cross-chain transactions; the other serves as a Relay Chain to implement cross-chain forwarding.

### 6.2. Computational Overhead

In GeoCross, cross-chain data privacy and authorized data access are achieved using ABE. This section evaluates this approach from both theoretical complexity and experimental performance. We focus on analyzing the impact of attribute quantity and access policy complexity on system performance, and compare our scheme with two typical CP-ABE schemes.

Theoretical Analysis: In traditional ABE, the computational complexity of KeyGen and Encryption is affected by both the length of the encrypted data and the complexity of the access policy. In GeoCross, we designed a constant-size plaintext encapsulation encryption, making the computational overhead primarily related to the access control policy. To analyze the impact of access policy complexity, we set the following system parameters. M: the number of attributes; m, n: the number of rows and columns in the access policy matrix (MSP); N: the length of the attribute. The computational complexity of each phase is shown in Table 1.

As shown in Figure 4, as the number of attributes increases, the computational overhead of KeyGen, Encryption, and Decryption also increases. To further verify whether this computational overhead satisfies the requirements, we conducted an experimental performance analysis.

The experiment fixed the data length (1 kB) and gradually increased the attribute set size k=10,20,…,100. The average computation time of the KeyGen, Encryption, and Decryption was tested. The experimental results are shown in Figure 5.

The results show that as the number of attributes increases, the computational overhead of each stage showed a linear upward trend. However, the growth rate of GeoCross in Encryption and Decryption operations is significantly lower than [45,46]. During the Encryption stage, the encryption time of GeoCross increased with the number of attributes, but the overall time consumption was reduced about 30% and 40% compared to [45] and [46], respectively. During the Decryption stage, GeoCross’s decryption process exhibits a significant advantage when the number of attributes is large. These experimental results are consistent with the theoretical analysis.

### 6.3. Communication Overhead

In GeoCross, cross-chain communication overhead comes from two aspects: the cross-chain overhead caused by granting authorization through the relay blockchain, and the overhead of establishing cross-chain proxies and relay forwarding. To obtain more realistic evaluation results, we simulated the cross-chain interaction process in a multi-chain environment to evaluate the cross-chain performance.

We analyzed the communication overhead incurred by cross-chain authorization through the Relay Chain. In GeoCross, cross-chain authorization is accomplished by collaborative blockchains using cross-chain contracts. The upload function and access request function within these contracts are key functions for cross-chain interaction and the source of communication overhead and latency. upload function is an invoke operation, requiring an execute-order-validate process, which increases latency in cross-chain communication and limits the efficiency of cross-chain authorization. This means the throughput of upload function determines the lower bound on cross-chain communication. The access request function does not require ordering, their throughput and latency are less affected by performance, determining the upper bound on cross-chain communication. Experimental results are shown in Figure 6. For the upload function, when the sending rate is below 200 tps, the Relay Chain’s throughput increases linearly with the sending rate, with average latency between 100 and 300 ms. When the sending rate exceeds 200 tps, throughput stabilizes at this level, while latency increases, indicating that the system is approaching its threshold. For access request function, the Relay Chain does not require consensus, resulting in better throughput performance than upload function. When the sending rate is within the range of 0–400 tps, the throughput maintains a linear relationship with the sending rate, with an average latency of approximately 10 ms. When the rate exceeds 400 tps, the system latency increases rapidly, the performance reaches a bottleneck. GeoCross offers an average latency of 10–20 ms for cross-chain read operations and 100–300 ms for cross-chain write operations, which satisfies the non-real-time but fast-response business requirements in geological operations.

To further verify the performance of cross-chain data sharing, we set up two types of cross-chain data sharing scenarios: one-to-one unidirectional and many-to-one unidirectional. The cross-chain transaction load was set to 500, 1000, and 2000 transactions, and the sending rate range was 50–450 tps (the system reached the performance threshold when it exceeded 400 tps). We tested the actual performance of the GeoCross. The experimental results are shown in Figure 7.

One-to-one unidirectional. A Requester Chain (Provincial Bureau Chain) was designed to initiate access requests to the Owner Chain (Geoscience Institute Chain). The experimental results are shown in Figure 7. Cross-chain transaction throughput reached a system bottleneck at around 390 tps. When the sending rate was below 400 tps, the throughput increased linearly with the rate, slightly decreasing after exceeding the threshold. Cross-chain latency was high at low throughput levels, gradually decreasing as throughput increased. When the sending rate was too high, latency increased significantly. During the cross-chain requests, the Provincial Bureau Chain requests data from the Geoscience Institute Chain to perform write operations, resulting in higher CPU usage than the Geoscience Institute Chain requires for read operations. Conversely, the Geoscience Institute Chain’s memory usage increases when performing read operations. Overall memory and CPU usage remain stable as throughput increases, indicating manageable system resource consumption.

Many-to-one unidirectional. Multiple Requester Chains (Povincial Bureau Chains) were designed to simultaneously initiate access requests to the Owner Chain (Geoscience Institute Chain). The experimental results are shown in Figure 8. When the sending rate reached 400 tps, the system throughput was about 1550 tps, demonstrating the architecture’s excellent scalability with multi-chain access. As throughput increased, cross-chain relay latency stabilized, but increased after exceeding a threshold. Resource consumption for nodes in multi-requester chains was similar to the single chain. However, the Geoscience Institute Chain needs to respond to multiple requests simultaneously, increasing its memory and CPU usage and becoming one of the performance bottlenecks.

To evaluate the communication performance of the proposed scheme, we further designed a relay node-based cross-chain interaction mechanism and incorporated two mainstream cross-chain frameworks, WeCross [22] and BitXHub [19], for comparative analysis. Similar to our design, the relay-node approach also requires reading and writing operations on functional contracts. However, its key limitation lies in the inability of relay nodes to persist with cross-chain transaction data; thus, cross-chain communication must be executed in real time. As a result, the performance ceiling of relay-based cross-chain interaction is determined by write-operation throughput. Due to the higher network complexity and consensus overhead of the service chain, the write throughput is approximately 140 tps. Using identical data payloads for cross-chain interaction, the performance comparison of the four schemes is shown in Figure 9. The results indicate that GeoCross achieves a higher performance upper bound than the mature cross-chain solutions WeCross and BitXHub, as well as the relay node scheme. Moreover, its lower-bound performance also exceeds that of BitXHub and the relay-node approach. Therefore, we conclude that the proposed scheme is capable of meeting the requirements of geological data sharing.

To verify the architecture’s scalability, we further conducted experiments with increasing numbers of requester chains (2–20 chains). The results are shown in Figure 10. When the number of requester chains ranged from 2 to 16, the system throughput increased linearly. However, when the number of requester chains exceeded 16, the throughput decreased by approximately 20% due to the bandwidth limitations in Relay Chain. In multi-chain access scenarios, GeoCross can achieve a total cross-chain throughput of approximately 1550 tps, which can satisfy the business scenarios where multiple provincial geological bureaus and research institutes simultaneously access cross-chain data.

Experimental results demonstrate that GeoCross exhibits excellent performance stability and scalability in both one-to-one and many-to-one cross-chain data sharing scenarios. The system maintains high throughput and low latency under high concurrency conditions. The system is capable of satisfying the needs of large-scale geological data sharing scenarios.

## 7. Conclusions

This paper proposes GeoCross, a secure and privacy-preserving geological data sharing scheme supporting cross-heterogeneous blockchains. By coordinating hierarchical authorization, privacy-aware verification, and trusted relay selection, this scheme demonstrates that distributed geological data can be shared without disclosing sensitive information or relying on centralized trust. The system can meet the practical needs of information exchange between geological systems and supports controlled collaboration in multi-chain environments. Security analysis and experimental evaluation show that GeoCross maintains stable performance under various workloads and real-world geological data sharing environments, making it suitable for practical applications such as inter-institutional data collaboration.

However, GeoCross still has some limitations. It still relies on trusted initialization involving a central authority and property authorities, which may limit large-scale deployment. Furthermore, while Groth16 ensures efficient verification, its proof generation overhead and fixed circuit structure reduce flexibility in handling heterogeneous or dynamic geological data. Future work will explore decentralized or threshold-based authorization structures, scalable or transparent zero-knowledge proof frameworks, and large-scale evaluation in real-world geological information systems. These improvements aim to further enhance GeoCross’s scalability, adaptability, and trust decentralization capabilities, enabling its wider application in complex, data-intensive geological environments.

## Figures and Tables

**Figure 1 sensors-25-07625-f001:**
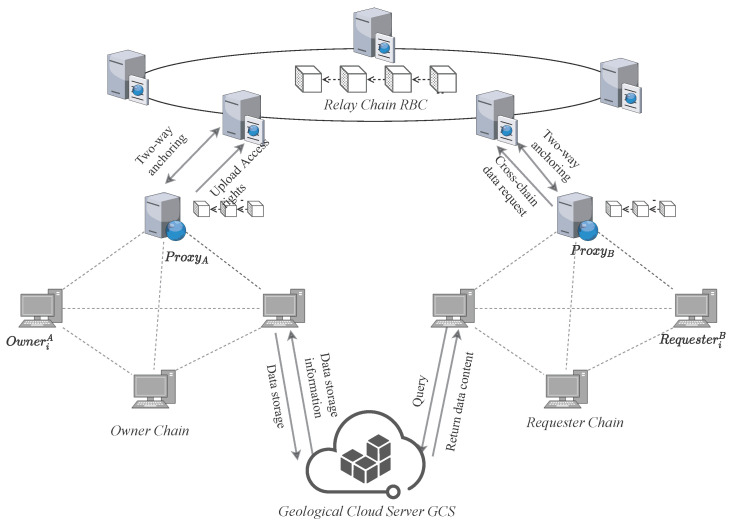
System architecture of GeoCross.

**Figure 2 sensors-25-07625-f002:**
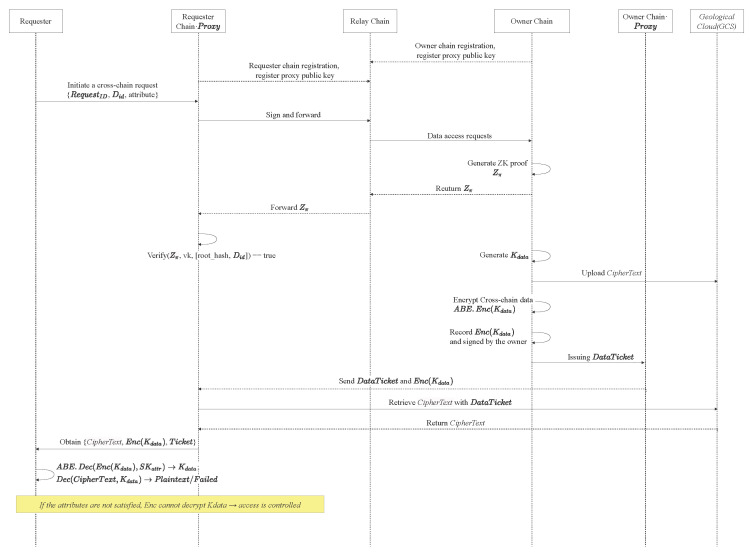
The overall process of cross-chain data sharing in GeoCross.

**Figure 3 sensors-25-07625-f003:**
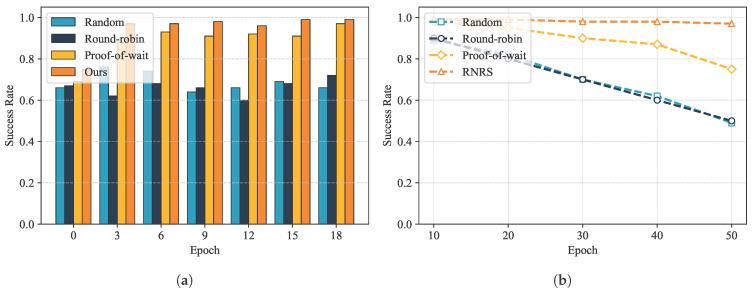
(**a**) The change in success rate with increasing Epochs; (**b**) the change in success rate with increasing proportion of malicious nodes.

**Figure 4 sensors-25-07625-f004:**
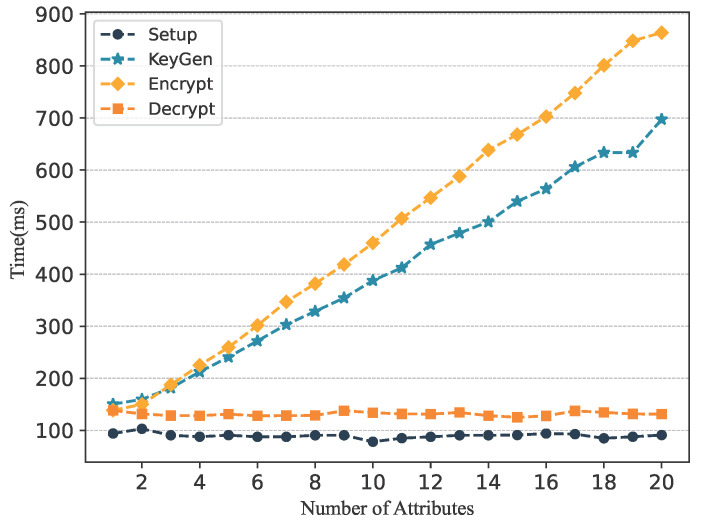
The performance of ABE is affected by the number of attributes.

**Figure 5 sensors-25-07625-f005:**
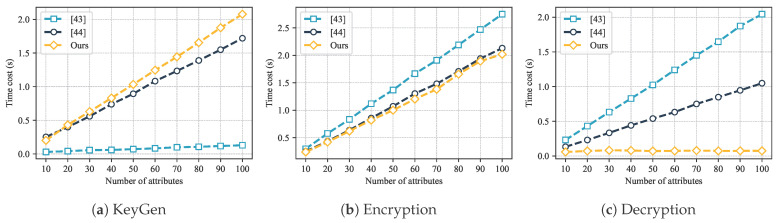
Comparison of computational overhead of different ABE schemes: (**a**) KeyGen; (**b**) Encryption; (**c**) Decryption [43,44].

**Figure 6 sensors-25-07625-f006:**
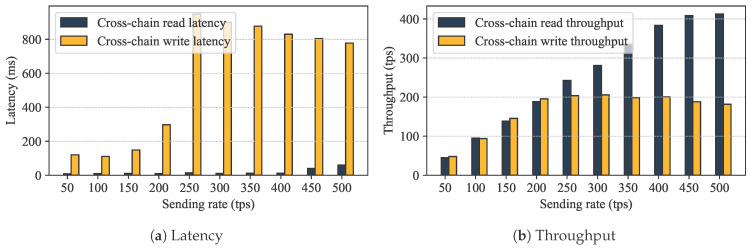
Cross-chain communication performance evaluation: (**a**) latency; (**b**) throughput.

**Figure 7 sensors-25-07625-f007:**
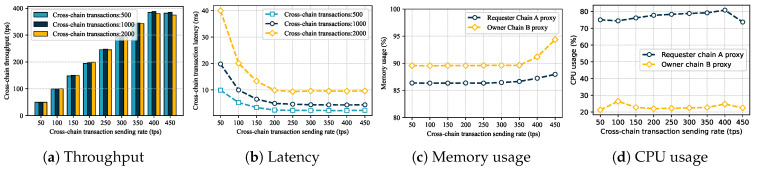
Cross-chain data sharing performance evaluation.

**Figure 8 sensors-25-07625-f008:**
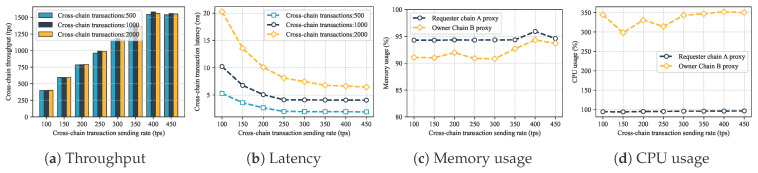
Cross-chain data sharing performance evaluation.

**Figure 9 sensors-25-07625-f009:**
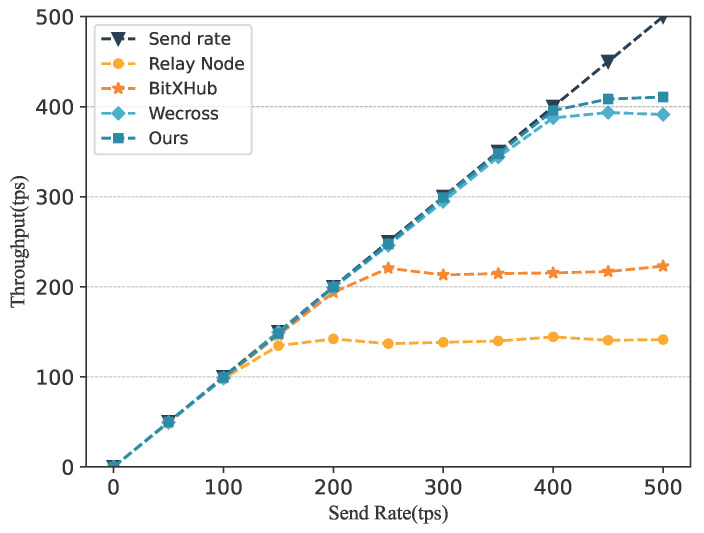
The comparison of different cross-chain schemes on throughput.

**Figure 10 sensors-25-07625-f010:**
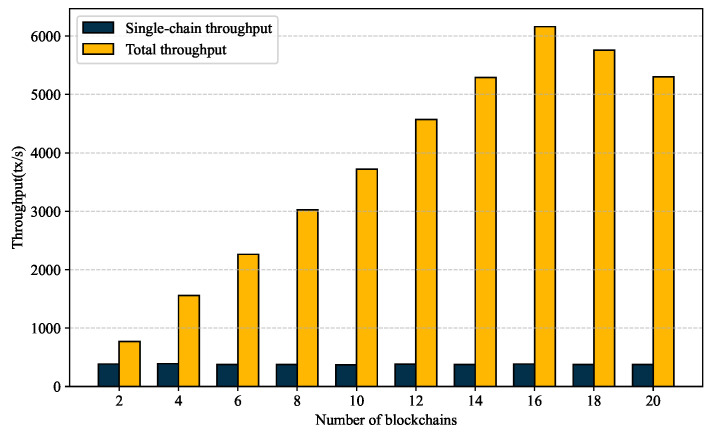
Impact of the number of requester chains on throughput.

**Table 1 sensors-25-07625-t001:** Comparison of the number of operations in different encryption algorithms.

Scheme	KeyGen						Encryption						Decryption		
	G			H			G			H			G	H	
	**MUL**	**EXP**	**Hash**	**MUL**	**EXP**	**Hash**	**MUL**	**EXP**	**Hash**	**MUL**	**EXP**	**Hash**	**MUL**		**Pair**
This	8M + 9	9M + 9	6M + 6	-	3	-	12mn+6m	6m	6m+6n	-	3	-	6N + 1	-	6
Waters2011 [43]	1	M+1	-	-	1	-	*m*	2m	-	-	m+1	-	N	-	N + 2
bethencourt2007 [44]	M + 1	M + 2	M	-	M	-	-	*m*	*m*		m+1	-	-	-	2N + 1

## Data Availability

The original contributions presented in this study are included in the article. Further inquiries can be directed to the corresponding authors.

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
