# Peer review of "GeoCross: A Privacy-Preserving and Fine-Grained Authorization Scheme for Cross-Chain Geological Data Sharing"

_sensors, 2025, doi:10.3390/s25247625_

Round 1
Reviewer 1 Report
Comments and Suggestions for Authors
This manuscript introduces GeoCross, a cross-chain scheme for sharing geological data. GeoCross combines hierarchical privacy protection with zero-knowledge verification to solve the challenges of inadequate privacy protection and trust deficiency among geological blockchains. The security proofs demonstrate that the proposed scheme satisfies adaptive semantic security under the random oracle model. Experimental results indicate that, compared to existing schemes, the proposed approach exhibits advantages in high efficiency in cross-chain sharing. The questions and suggestions for authors are as follows. 1. The abstract is not properly written. The abstract should implicitly include the purpose/problem, methods, and numerical findings. It requires comprehensive improvement. 2. The main contributions are exaggerated and not written in depth. 3. In the RELATED WORK section, the authors did not provide sufficient critique of all the included research, making the research gap unclear. 4. It is recommended to include a diagram that illustrates the proposed methodology. 5. Analyzing performance results is not enough. 6. The authors did not provide comparisons (references) between their findings and the findings of the most closely related research listed in the Related Works section. 7. Originality: The authors should prove the novelty and originality of the presented work, as a lot of studies have already been published in the literature. 8. Duplication: Remove any duplication between the abstract and the conclusion. 9. What are the limitations of this research? 10. The conclusion of the study is unclear in the conclusion section.
Reviewer 2 Report
Comments and Suggestions for Authors
Introduction
The problem of secured data storage in cloud systems has to be explain with more detail because you affirm that current cloud storage is not secure nor private (line 30-32). But if you do not trust on data cloud storage, how are you storing your blockchain? Apart from the fact that blockchain stores information in a traceable and unmodifiable way, the storage systems are equally vulnerable to attacks, because and attacker that access your system can corrupt the chain or delete it even though they are distributed because they also need network access. Moreover, information in cloud storage can be also ciphered, so it can remain private after and attack and the logs can be protected too. These facts should be better explained in this section.
Related work
This section needs a deeper analysis of other approaches and a detailed explanation of their flaws of disadvantages compared to the authors’ proposal. Moreover, the date of the only three referenced papers (29,30, 31) is slightly old. Please provide more recent solutions to this problem, comment and compare all of them with more detail. It is possible that you have to mix introduction section and related work section if order to explain cross-chain schemes in the related work.
Section 3
Line 177 “We assume that the central authority and attribute authorities are fully trustworthy” Why? How? All network infrastructures are susceptible to attack.
Section 4
Please, comment how your proposal is protected from the “Man in the middle” attack in the setup phase.
Again, in the process of communicating the public key of a proxy (paragraph at line 265) how is made this process? What about “man-in-the-middle” attacks here?
Section 6
For section 6.2 and 6.3, the results must be compared with other similar approaches to test the efficiency of the system. Results without comparison lack of significance.
Minor issues
Try to not use abbreviatures in the abstract. Define them in the other sections the first time you used them.
Line 14: typo – “the its”
Line 73: Please add reference to Groth16.
Lines 86-100: the bullet list appears to have undesirable blank spaces at line beginning.
Line 146: typo “registration registration”
Line 594: BSW and Waters need references.
Round 2
Reviewer 2 Report
Comments and Suggestions for Authors
Minor changes:
Line 139: "... that blockchain as significant...": as should be has.
Lines 154: please revise if "investigating" is appropriate
